# GC-MS and Sensory Analysis of Aqueous Extracts of Monovarietal American Hops, Produced Using the Synergy Pure™ Extraction Technique

**DOI:** 10.3390/foods13152454

**Published:** 2024-08-03

**Authors:** Gianluca Tripodi, Alessio Cappelli, Marta Ferluga, Giovanna Dima, Mauro Zaninelli

**Affiliations:** Department of Human Sciences and Promoting of the Quality of Life, San Raffaele Telematic University Rome, Via Val Cannuta 247, 00166 Rome, Italy; alessio.cappelli@uniroma5.it (A.C.); mauro.zaninelli@uniroma5.it (M.Z.)

**Keywords:** *Humulus lupulus* L., aqueous extracts, aroma profile, volatile profile, American variety of hops, Citra, Mosaic, Centennial, Chinook

## Abstract

Aqueous extracts from four different American hops varieties (Mosaic, Chinook, Citra, and Centennial) were produced using the Synergy Pure technique, an innovative extraction distillation process developed by Synergy Flavours, a global specialist in the manufacturing of flavors, extracts and essences. This process is able to preserve and maximize the aromatic characteristics without increasing the bitterness of the final product. Therefore, the aim of this work is to identify the volatile and sensory characteristics of these extracts, with the additional aim to assess their suitability for brewing. GC-MS and sensory analyses were carried out on the four different aqueous extracts. The results highlighted the presence of 33 volatile compounds in a quantity that allowed us to identify the characteristics of the varieties under investigation in each extract. Sensory analysis was carried out using the expert sensory profiling technique. The results regarding the olfactory analysis of the extracts allowed us to define the aroma profiles of the four extracts, highlighting their strong correspondence with the characteristic of the varieties under investigation. Finally, the results showed that the aqueous extracts produced using the Synergy Pure extraction technique had a richer aroma profile, highlighting its higher aptitude in beer production.

## 1. Introduction

Hop (*Humulus lupulus* L.) is a perennial climbing plant that grows in areas with a moderate climate [1]; it belongs to the Cannabaceae family of the order Urticales [2,3] and is widely used in the brewing industry for the properties of its inflorescences, particularly lupulin [4].

The sensory characteristics, flavor and aroma of hops are mainly attributable to several components contained in the inflorescences. These compounds belong mainly to three categories: resins (hard and soft), which are present in a percentage between 3% and 5%; polyphenols, approximately 4%; and essential oils, between 0.5% and 3% [1].

The soft resins contain α- and β-acids that, during the boiling process, alter their chemical structures, forming iso-α-acids in cis and trans forms, respectively. These products are important contributors to the bitter flavor and the preservation of beer [5]. For this reason, hop resins mainly provide to beer its typical bitter taste, while essential oils contribute to the complexity of the aroma [6].

The compounds in the essential oils of hops belong to several chemical classes and might be present in different percentages, according to the hop variety. Moreover, the total essential oil content and composition is influenced by several factors, such as the age of the plant, soil and climate factors, environmental quality, harvesting ripeness, drying level, and oxidation and storage conditions [7].

Various studies have suggested that more than 1000 compounds may be present in hops’ essential oils [8,9]; these compounds can be attributed to three classes (hydrocarbons, oxygenated compounds, and sulfur-containing compounds) [10,11]. Therefore, there is varietal uniformity in the chemical composition of the essential oils, but their proportions depend on the hop variety [10,11].

With respect to the monoterpenic alcohols, a literature review highlighted that linalool has an essential role in beer aroma formation. In particular, the latter compound provides a floral aroma with sensory notes of lavender, with a perception threshold of 1–3 µg/L in water, which is very low compared with other hop compounds [12]. Furthermore, β-citronellol significantly contributes to the creation of the beer flavor; β-citronellol provides an aroma associated with citrus and lime in beer, with a perception threshold of 8–9 µg/L. Another important compound that influences the beer flavor is geraniol; this is characterized by a perception threshold of 4–7 µg/L and a floral character [12]. Interestingly, the coexistence of these three monoterpenic alcohols enables a citrusy aroma profile that is reminiscent of lime [12].

With respect to the thiol group, researchers have focused their studies on substances belonging to this class of compounds, like 3-methyl-2-butene-1-thiol, that are considered unpleasant to taste. However, since the beginning of the 2000s, research has turned its attention to studying thiols associated with fruity aromas and their precursors, present in both malt and hops [13,14].

Esters are another group of compounds responsible for hops’ varietal differentiation. Among these, the isobutyric esters of branched-chain alcohols, such as isobutyl, isobutyrate, isoamyl isobutyrate, and 2-methyl-butyl-isobutyrate, are the most widely investigated in the literature [15]. Sensory analysis studies have correlated isobutyric esters with a fruity aroma associated with green, apple, and apricot scents; among the aforementioned isobutyric esters, 2-methylbutylisobutyrate stands out in terms of quantity, with a perception threshold of 78 µg/L in beer [7].

The literature review reported only a few studies aiming to assess the volatile composition of hop extracts, often examining only a limited number of compounds. Moreover, no paper in the literature has evaluated the volatile composition of aqueous hop extracts. 

With respect to the techniques used to produce hop extracts, the literature review highlighted that steam distillation and supercritical fluid extraction with CO_2_ are considered the standard methods by both the American Society of Brewing Chemists and the European Brewery Convention [10,16]. However, this traditional technique presents several disadvantages, like the need for large sample amounts, the high time consumption, the impossibility to be automatized, and the lower quality in terms of aroma due to the degradation of non-volatile compounds and the formation of defects [10,16]. The latter disadvantage, in particular, has driven researchers and brewmasters to search for new, innovative methods. 

Among these innovative methods, the one developed by Synergy Flavours (Trieste, Italy) seems to be one of the most interesting. The products created using this approach are trade-marked with the branding Synergy Pure™. This technique is based on an innovative patented distillation process that is able to preserve and maximize the aromatic characteristics of the hop extract without increasing the bitterness of the final product. This is mainly due to two reasons: the absence of resins in the aqueous extracts and the limited degradation of the volatile compounds, which are preserved in the final products. Consequently, the aqueous extract should be highly concentrated in aroma and not bitter at all. Moreover, the Synergy Pure™ extraction system only uses water as the solvent, highlighting a significant reduction in environmental impacts; this can provide an additional advantage on the market, given the increasing interest of consumers in eco-friendly food products. 

In this study, we used four American hop varieties. The United States is now the world largest producer of hops, with more than half of the world’s production each year [17]. The American hop varieties are characterized by greater pest resistance [3] and aromatic intensity compared to European varieties [18]. 

Regarding the varieties analyzed in this research, the Citra hop was created in the United States in 1990 through a series of crosses between unregistered hop varieties and an American wild hop [17]. Its complex varietal profile is characterized by citrusy aromas, including grapefruit, lime, and orange, as well as woody and herbaceous notes [19]. These sensory characteristics have made this one of the most popular hop varieties in the United States in recent years [17]. 

The Centennial variety takes its name from the celebration of Washington State’s 100th anniversary in 1989. Together with the Cascade and Columbus varieties, it forms the “3C group” (i.e., the hops that best represent the essence of American varieties in terms of aroma). They have also contributed to the “craft beer revolution” movement in the USA. In addition to their woody aromas, this variety includes the spicy aromas of spruce, tonka bean, estragon, and aniseed, combined with fruity secondary notes of raspberry and exotic aromas such as lychee and pineapple [17].

The Chinook hop took its name from a Native American tribe in the region around Washington. This variety is one of the most popular in the American craft beer scene. It is currently a popular variety on the market for its flavor profile and resilient adaptability to the Mid-Atlantic environment. Beers brewed with this hop have a well-balanced flavor characterized by floral rose and fruity aromas, particularly tropical and green ones, as well as resinous and balsamic notes [17].

Mosaic is a newly developed American hop variety that has been on the market since 2012. This name is related to its “artistic” complexity in terms of aromas and flavors. It is also known for its predominantly citrusy and tropical aromas, including pineapple, mango, and passion fruit [17].

Sensory analysis can support analytical studies, like GC-MS, in understanding the composition of a hop varietal’s aroma and the chemical–sensory interactions of its components [6,10,11,19,20,21]. In the present work, the expert sensory profiling technique has been coupled with GC-MS to obtain a complete and extensive assessment of the aqueous extracts’ compositions. 

Therefore, the first aim of this work is to characterize, for the first time, the total volatile fractions and the sensory profiles of aqueous extracts produced with the Synergy Pure™ extraction system. The second aim of this work is to compare these extracts with traditional extracts present in the literature to define those that are more suitable to be used in beer production.

## 2. Materials and Methods

### 2.1. Raw Materials and Preparation

The cultivars analyzed were Mosaic, Chinook, Citra, and Centennial. All cultivars were purchased from the Yakima Chief Hops Company (Yakima, WA, USA) and stored at 6 °C until processing. The extraction process involved the use of hop inflorescences, without any pre-treatment, such as drying or crushing. The purchase of the hops and the extraction process were carried out by Synergy Flavours (Trieste, Italy).

### 2.2. Chemical Standards and Reagents

Toluene (ACS grade, ≥99.5%), which was not water-soluble, diethyl ether (ACS grade ≥99.0%), and the analytical standards (ethyl-nonanoate) used for GC-MS analysis were purchased from Sigma-Aldrich (St. Louis, MO, USA).

### 2.3. Aqueous Extract Production

The aqueous extracts were produced by Synergy Flavours (Trieste, Italy) using the patented Synergy Pure™ distillation process. This technique involves vacuum extraction from hop pellets without the use of solvents.

In the extraction process, 30.0 g of hop pellets was weighed in a 2 L round-bottomed flask. Then, 970 mL of water was added and the mixture was stirred for 15 min to obtain a homogeneous slurry. The slurry was distilled on a rotary evaporator, with a bath temperature of 95 °C and at 475 mbar internal pressure. The slurry was distilled until a residue of 50.0 g was obtained at the bottom of the flask. This final slurry was used to prepare samples for the following analyses.

### 2.4. Sample Preparation

For the GC-MS analysis, 1 g of hop extract was placed inside an 8 mL vial, adding 4 g of water, 25 µL of 0.5% *w*/*v* internal standard solution, and 1.60 mL of diethyl ether. Ethyl nonanoate was used as an internal standard at a concentration of 1 mg/mL. The sample was mixed via vortex shaking for 1 min and then centrifuged at 4500 rpm for 3 min. Then, 1 mL of supernatant was transferred to a 1.5 mL gas chromatography vial.

For the sensory analysis, 1.5 g of extract was used in 100 g of demineralized water at a room temperature of about 25 °C. Samples were randomized using a three-digit code and presented in capped tasting glasses with clear, odorless plastic caps.

For each variety, aqueous extracts were produced in three replicates using 3 different batches, for a total of 12 samples.

### 2.5. GC-MS Analyses

GC-MS analyses were conducted using an Agilent GC 6850 gas chromatograph coupled with an Agilent MS 5975 mass spectrometer (Agilent, Cernusco Sul Naviglio Milan, Italy). An Agilent J&W DB-FFAP nitroterephthalic-acid-modified polyethylene glycol (PEG) column (length, 30 m; inner diameter, 0.25 mm; stationary-phase thickness, 0.25 µm) was used (Santa Clara, CA, USA). 

Direct-type injection was carried out using an autosampler with an injector temperature of 250 °C; a split ratio of 5:1 was used for the injection mode.

The carrier gas sed was helium, with constant pressure of 12 psi and with a temperature schedule that included an initial temperature of 60 °C for 5 min, a ramp of 4 °C/min up to 240 °C, and 10 min at 240 °C; the total run time of the GC analysis was 60 min. The transfer line temperature was 250 °C.

Data acquisition was performed in scan mode (*m/z* 30–300) with a scan event time of 0.30 s. The data processing software employed was Agilent ChemStation version B.04.03. 

Qualitative analysis was performed by comparing the mass spectra of individual peaks with the available databases (NIST, Wiley, Agilent Flavours Database, and an internal database), by comparing both the linear retention indexes (LRI) and data from the literature. The LRIs were calculated according to Cincotta et al. [22]. 

Quantitative determination was performed with the internal standard technique using ethyl-nonanoate and individual substance response factors. Aliquots of ethyl-nonanoate aqueous solution (1 mg/mL) were added to all samples. 

The analysis of the twelve samples obtained from the extraction process described in Section 2.3 was carried out in three replicates (a total of 36 samples were analyzed using GC-MS, with 12 samples analyzed in triplicate).

### 2.6. Sensory Analysis 

The sensory analysis was carried out on the aqueous extracts using the expert sensory profiling technique, according to the guidelines defined in the ISO standard for panel definition [23] and defining training methods for judges [23,24]. This descriptive technique involves a qualitative phase aimed at generating descriptors considered part of the lexicon and a quantitative phase aiming to evaluate the intensity of the chosen descriptors [25,26,27].

The sensory panel consisted of 10 trained people (5 females and 5 males) with ages between 30 and 55 years. The judges had worked for at least two years in the tasting and evaluation of plant extracts (including hop extracts). The participants were asked to refrain from smoking, eating, or drinking (excluding water) in the three hours before the test sessions.

The panelists gave their written consent after receiving full information about the sensory test. The subjects experienced no risk as a result of the sensory test. 

The panel defined 8 olfactory descriptors: “citrus”, which indicated citrus notes such as lime, grapefruit, orange, tangerine, and others; “sweet fruit”, which included both stone fruit and all fruit with particularly sweet and ripe notes, such as peach, grape, and red apple; “tropical fruit”, such as mango, pineapple, papaya, guava, and others; “green fruit”, such as kiwifruit, green apple, and others; “floral”, such as rose, violet, orange blossom, and others; “herbaceous”, such as cut grass and similar aromas; “spicy/woody”, such as pepper, cinnamon, oak, and others; and “resinous/pine”, such as conifer resin and balsamic notes. The intensities of the olfactory descriptors were defined for each sample using a scale from 1 to 10, in which “1” indicated not present at all and “10” indicated extremely present. During each evaluation session, the judges analyzed two hop extracts, restoring the palate with water between each tasting. All extracts were evaluated in three replicates, for a total of 36 sensory assessments.

### 2.7. Statistical Analysis

The XLStat software, version 2014.5.03 (Addinsoft Incorporated, New York, NY, USA), was used to statistically evaluate the GC-MS and sensory data. 

The one-way ANOVA test and principal component analysis (PCA) were performed to check for significance between the mean values of the cultivars analyzed. A significance level of *p* < 0.05 was considered statistically significant.

## 3. Results and Discussion

### 3.1. GC-MS Analyses

Table 1 summarizes the results of the four monovarietal aqueous extracts. The results of the GC-MS analysis identified 33 different volatile compounds. In particular, Table 1 shows the identified compounds, the retention times, the linear retention index, the amounts expressed in parts per million (ppm), and the odor descriptors. 

The compounds identified belonged to the terpene, sesquiterpene, ester, ketone, acid, alcohol, and sulfur compound classes. 

The main compounds identified were myrcene (fresh and balsamic), α-humulene (earthy, woody, and spicy), and β-caryophyllene (woody and spicy), with the following percentages: 50.18%, 20.20%, and 10.91% for the Citra extract; 52.90%, 17.95%, and 8.83% for the Centennial extract; 24.72%, 37.66%, and 16.36% for the Chinook extract; and 43.22%, 22.64%, and 7.88% for the Mosaic extract, respectively. Only the Chinook variety had α-humulene as its main compound, while the other varieties had myrcene as their main compound, albeit with different percentages. From an organoleptic point of view, myrcene is associated with fresh, balsamic notes; α-humulene with spicy and woody notes; and, finally, β-caryophyllene with spicy and woody notes [11,28]. Despite their high concentrations in hops, they are often only transferred into beer in small amounts due to their non-polar character. Thus, they are considered to contribute only to a minor degree to the aroma and flavor in the final beer. However, they play an important role as the precursors of compounds that contribute to the aroma and flavor of “noble hops” or “kettle hops” in beer [29].

Monoterpene alcohols are generally biosynthetic products related to the biosynthesis of myrcene. Linalool and geraniol were identified in all samples; these compounds have been found to contribute to different fruity and floral dimensions of the hop aroma [11]. Linalool showed high amounts in the Citra variety samples (47.87 ppm), about two times higher than in the Mosaic samples (23.63 ppm) and three times higher than in the Chinook samples (17.67 ppm). Geraniol was lowest in the Citra variety (9.85 ppm) and highest in the Centennial variety (52.23 ppm). Linalool, well known as a major indicator of the hopped beer flavor, has a floral, lavender-type flavor, and its perceptive threshold is very low (1–3 µg/L). Geraniol has a rose petal flavor and a slightly higher threshold (4–7 µg/L); however, it seems to be able to contribute to the beer flavor. These monoterpenes have been identified in several varieties of both European and American hops by other authors, supporting the obtained results [10,11,12,30]. However, in the samples analyzed, the linalool content was 2–3% of the total, which was higher than the approximately 1% observed with the other extraction techniques. Furthermore, it has been found that the concentration decreases rapidly during the brewing process [31,32].

Linalool and geraniol have been found to interact with compounds from other chemical classes, such as fermentation by-products, to increase the floral aroma characteristics of the final beer [9,33]. Thus, a higher concentration of them in the hop extract could improve the interaction with other compounds by amplifying the floral notes.

In addition to the differences discussed for the three basic compounds (myrcene, α-humulene, and β-caryophyllene), the number of terpenic compounds identified in the samples differed mainly in the following aspects: α-pinene and linalool were higher in the Citra variety and geraniol in the Centennial variety; β-pinene was higher in the Centennial and Citra samples; and ylangene, α-cubebene, γ-muurolene, and valencene were higher in the Chinook samples. Given this evaluation, it was possible to affirm that Mosaic was the least represented from a varietal point of view, while the Chinook variety had high terpene and sesquiterpene values, although with relatively low overall amounts of volatiles.

Hops mainly contain modest amounts of esters with branched-chain structures [10]. In the samples analyzed, the main esters identified were isoamyl isobutyrate (fruity, apple-like), methyl 4-decenoate (waxy, milky, green), and geranyl isobutyrate (sweet floral, fruity, green). The amounts of these compounds differed between the varieties analyzed. Isoamyl isobutyrate showed higher amounts in the Citra variety (47.75 ppm), about twice as high as in the Chinook (26.61 ppm) and Centennial (25.32 ppm) varieties. Methyl 4-decenoate also showed an amount of 48.68 ppm in the Citra samples, which was almost ten times higher than in the Chinook samples (5.43 ppm). Only geranyl isobutyrate was more present in the Centennial variety samples. Esters, and particularly methyl esters, contribute to the aroma and flavor of hops in beer due to their low threshold concentrations [34,35]. Isobutyric esters consist of isobutyric acid and branched-chain alcohols; these structures are derived from branched-chain fatty acids (isobutyric acid, isovaleric acid, and 2-methylbutyric acid) and branched-chain amino acids (valine, leucine, and isoleucine). In brewing, these structures can be traced to the side chains of hop bitter acids (α-acids, iso-α-acids, and β-acids) [11,36]. A significant amount of esters, present in the volatile fraction of hops, is hydrolyzed by yeast or trans-esterified during brewing; however, the conjugated acid esters resist hydrolysis and are transferred to the final beer in their original form [37,38].

Three ketones were also identified in the samples: 2-nonanone, 2-undecanone, and 2-tridecanone. The amounts of these compounds were usually in the range of units. Exceptions were the amounts of 2-undecanone in the Citra (44.74 ppm) and Mosaic (34.02 ppm) samples and 2-tridecanone also in the Mosaic variety (13.88 ppm). Dietz et al. (2020) and Eyres et al. (2016) [11,34] reported the presence of an isomeric series of methyl ketones in essential hop oil, the main compound of which was 2-undecanone, consistent with our results. It has been found that the sensory profiles of ketones depend strongly on their molecular weights. In fact, the higher the molecular weight, the more the fruity aroma is transformed into a floral aroma [9].

The oxygenated sesquiterpenoids identified in the aqueous extracts analyzed were β-caryophyllene oxide and humulene epoxide, which are considered positive in beer aroma development (described with woody and spicy odor notes) [39]. These compounds showed lower amounts in the Chinook samples: 1.64 ppm for β-caryophyllene oxide and 5.50 ppm for humulene epoxide. The highest amounts were found in the Citra samples, with 14.37 ppm and 16.45 ppm, respectively.

These compounds result from the autoxidation, subsequent hydrolysis, and rearrangement of sesquiterpene hydrocarbons, which usually occur during hop storage [11,34]. Goiris et al. (2002) and Praet et al. (2014) have attributed the spicy and herbaceous notes of the hop aroma to these compounds in the essential oil of raw hops [40,41]. 

The sulfur compounds identified (dimethyl sulfide and dimethyl disulfide) were present in small amounts in the samples analyzed. The highest amounts were found in the Mosaic variety (1.32 ppm). Dietz et al. (2020) reported that these compounds are present in small amounts in hops and beer, often at trace levels [11]. Dimethyl sulfide and dimethyl disulfide have a low odor threshold: 7.5 and 15 µg/L, respectively [11,42]. These compounds may be the product of S-methylcysteine sulfoxide degradation during the extraction process [34]. The organic sulfur volatile compounds have been identified with difficulty in the volatile fractions of hops and beer [42]. However, these compounds contribute to the aroma of hops in beer, to changing the perception of other aroma compounds, and to imparting “unpleasant” flavors to beer [42].

Linear medium-chain acids, such as octanoic acid and decanoic acid, were identified and quantified in all samples analyzed. The amounts of these compounds ranged from 0.10 ppm of octanoic acid in the Chinook samples to 4.74 ppm of decanoic acid in the Citra samples. Acids are usually associated with the degradation of α- and β-acids as hops age. However, these compounds have high odor thresholds: 4.5 and 1.5 mg/L, respectively [43]. Their presence is greatly influenced by the method of sample preparation and the extraction or distillation used to isolate the oil [34].

Examining the quantitative data of the tested aqueous extracts, it is possible to highlight that the Citra variety showed higher volatile compound content, while the Mosaic variety showed lower amounts. This difference was due to the lower amount in the latter variety of the three main compounds (myrcene, α-humulene, and β-caryophyllene), with values approximately 30% lower. Although they had similar total values, the Centennial and Chinook varieties were different in the general predominance of mycene and α-humulene, respectively. 

In detail, the aqueous extract of Citra hops mainly consists of terpenes, as does the essential oil of hops of this variety [13,44]; among these, the main ones in terms of quantity are myrcene, α-humulene, and β-caryophyllene, which are known to play an essential role in the hop aroma [45]. Considerable concentrations of esters such as methyl 4-decenoate (related to waxy, leathery, fruity, pear, milky, and green flavors) and isoamyl isobutyrate (fruity and apple-like flavors) are also present in this variety. Linalool accounts for a modest fraction of the total; despite this, it plays a key role in the aroma of Citra hops [46,47]. There were also considerable amounts of 2-undecanone (fruity, creamy, fat, iris, and floral flavors) and valencene (woody, citric, and citrusy flavors) in this variety.

The Centennial variety samples were characterized by high concentrations of myrcene, α-humulene, β-caryophyllene, geraniol (rose petal fragrance), and linalool (floral, woody, and fresh notes) [47,48], while the amounts of the esters, especially isoamyl isobutyrate and methyl 4-decenoate, were modest.

In addition to the three main volatile compounds found in all varieties, extracts from the Chinook hop variety were characterized by considerable amounts of terpenes and sesquiterpenes, such as valencene, associated with citrus notes; γ-murolene, linked to spicy and resinous notes; and α-cubebene, which imparts herbaceous notes [21,47]. 

The Mosaic variety had a lower number of volatile compounds, but the extracts were also characterized by ketones such as 2-undecanone, associated with herbaceous, fruity, creamy, and floral sensory hints, and 2-tridecanone, linked to fat, earthy, and herbaceous notes. There were also considerable quantities of isoamyl isobutyrate, related to fruity hints, sometimes reminiscent of apples. Both compounds are typical of the aroma and chemical components of this variety [44,49].

The volatile composition data were subjected to statistical principal component analysis. Figure 1 shows that component 1 was indexed as 92.09% and PC2 as 7.55%. However, it was in component 2 that the varieties were distinguished, influenced mainly by the three main volatile compounds: myrcene, α-humulene, and β-caryophyllene. As illustrated in Figure 1, the Chinook variety samples were in the positive quadrant for PC1 and PC2, characterized by the incidence of α-humulene and β-caryophyllene, while the remaining varieties in the positive quadrant for PC1 and the negative quadrant for PC2 were characterized by the incidence of myrcene.

### 3.2. Sensory Analysis 

The sensory analysis conducted by the panel allowed us to collect data regarding the intensity of the predefined attributes of the aqueous extracts obtained from the Chinook, Centennial, Mosaic, and Citra hop varieties. 

The sensory data were used to create an olfactory profile of the aqueous extracts, as shown in the spider plot of Figure 2. Overall, small variations were found in the individual judges’ perception values, confirming the importance of intensive evaluator training to limit the variability and increase the robustness of the results.

According to the sensory data (Figure 2), all aqueous extracts analyzed were characterized by notable spicy/woody notes typical of hops [36]. 

The aroma notes revealed agreed with the qualitative analysis, particularly with the abundance of compounds that confer this note, including α-humulene, β-caryophyllene, valencene, γ-muurolene, humulene epoxide, and β-caryophyllene oxide [19].

Comparing the sensory data of the monovarietal aqueous extracts, it is possible to observe that the Citra extract was characterized by higher intensities for descriptors such as green fruit, herbaceous, and spicy/woody and lower intensities for floral descriptors. In particular, the herbaceous note is attributable to compounds such as myrcene, linalool, 2-undecanone, methyl 4-decenoate, and isoamyl isobutyrate [18,44]. Conversely, the Centennial extract was characterized by citrus and sweet fruit notes and the least by floral notes. These notes are related to volatile compounds like geraniol, linalool, and geranyl isobutyrate [50]. The Chinook extract scored the highest values for tropical fruit and resinous notes; moreover, it had the lowest values for citrus notes. The spicy/woody notes can be attributed to the presence of various terpenes and sesquiterpenes found in these samples, as well as the notable presence of α-humulene [18]. Finally, the Mosaic extract had the highest citrus and floral notes and the lowest values with respect to tropical, green, and sweet fruity descriptors; it was mainly characterized by floral and citrus notes due to ketones and compounds such as isoamyl isobutyrate [43]. 

These sensory evaluations agreed with the sensory profiles released by the world’s leading hop producer, the Barth Haas Group “https://www.barthhaasx.com/hops-and-products/hop-varieties-overview (accessed on 1 March 2024)”.

The sensory data were further evaluated using PCA. The first two components affected the distribution of the samples by 42.12% in PC1 and 25.39% in PC2 (Figure 3). Figure 3 shows, in graphical form, the main olfactory descriptors that characterized the four monovarietal aqueous extracts. The Chinook variety was mainly associated with a resin/pine descriptor, corresponding to its main aroma; the Mosaic variety was associated with spicy/woody and citrus descriptors; the Citra variety was associated with herbaceous and floral descriptors, which represent the main olfactory traits associated with this variety; finally, the Centennial variety was associated with green, sweet, and tropical fruit descriptors. 

More precisely, the Citra variety extracts were characterized by higher herbaceous notes, attributable to compounds such as myrcene, linalool, 2-undecanone, methyl 4-decenoate, and isoamyl isobutyrate [18,44]. On the other hand, the Centennial variety samples had high floral notes related to volatile compounds like geraniol, linalool, and geranyl isobutyrate [50]. In addition to the spicy/woody notes, Chinook variety extracts were characterized by higher resin/pine notes, which can be attributed to the presence of various terpenes and sesquiterpenes found in these samples, as well as the notable presence of α-humulene [18]. In contrast, the Mosaic variety was mainly characterized by floral and citrus notes due to ketones and compounds such as isoamyl isobutyrate [43].

### 3.3. Comparison of the Volatile Characteristics of Synergy Pure Extracts by Distillation Extraction

The comparison of the data was not easy, because the volatile fraction is very variable according to the hop variety, and there are very few literature papers that examine their complete volatile compositions. 

Several studies have reported that the extraction technique can influence the quality and quantity of the volatile fraction [51,52]. 

However, comparing the data with Duarte et al. (2020), it was observed that the average percentage of myrcene content (42.76%) was significantly higher than reported for distillation (28.68%). The average β-caryophyllene content in the aqueous extracts (11.10%) was in agreement with the literature data (14.45%), while only the α-humulene content (24.61%) was much lower in our extracts compared to extracts with distillation (42.20%). The main monoterpene alcohols showed higher average data in the aqueous extracts. In particular, the values for linalool (2.59%) and geraniol (2.20%) were almost twice as high as those obtained by distillation, 1.57% and 1.24%, respectively. Comparing the content of minor sesquiterpene compounds, α-cubebene (0.90%) and γ-muurolene (1.89%) showed higher content compared to distillation extraction, at 0.08% and 1.51%, respectively [45]. 

Moreover, it was encouraging to observe that the amounts obtained through the Synergy Pure™ extraction technique were in agreement with new extraction techniques, currently under testing, such as ultrasound-assisted solvent extraction, which employs both hexane or dichloromethane [45]. 

The myrcene percentages obtained by ultrasonic techniques with hexane were 45.30% for Mosaic and 44.30% for Citra varieties, being respectively, lower than or in agreement with those found in the aqueous extracts (50.18% for Mosaic, 43.23% for Citra varieties).

The β-caryophyllene content obtained with the ultrasound technique, using hexane, was higher than in our samples. In fact, the content was 13.70% in the ultrasonic technique and 7.88% in the aqueous extracts for the Mosaic variety, while, for Citra, it was, respectively, 15.80% versus 10.91%.

Meanwhile, the percentage content of α-humulene was in agreement with that in ultrasound extracts. The aqueous extracts showed content of 20.20% for Citra and 22.64% for Mosaic varieties, while Duarte et al. reported percentages of 22.70 and 27.40, respectively.

Moreover, the linalool content, obtained by ultrasonic extraction techniques, confirmed the extractive superiority of the Synergy Pure™ technique. In fact, the percentage content of linalool was 1.34 for Mosaic and 2.09 for Citra, while, in the aqueous extracts, it was 3.12% and 2.54%, respectively [45].

## 4. Future Perspectives

Authors have shown that the volatile compounds of hops change during the brewing process. These transformations reduce the amounts of many compounds and lead to the formation of new ones [10,11,36,39,53].

Furthermore, the perception of hop-derived volatiles is influenced by the beer matrix in which they are found; this is due to various factors such as the pH, temperature, yeasts, alcohol content, phenolic compounds, etc. [54]. Dietz et al. (2020) suggest that volatile compounds have additive, synergistic, or configural processing behavior that causes changes in flavor perception [11].

These prerequisites make further studies necessary on the use of aqueous hop extracts during brewing, to verify their impact on the final beer quality and assess their stability, their alterations, and the sensory characteristics perceived and possibly appreciated. 

Synergy Flavours describes aqueous hop extracts as free of bittering substances (hop acids, polyphenols), making their use as bitter hops unnecessary. However, the aqueous extracts obtained from the Synergy Pure technique could be used as aroma hops by adding them to green and bright beer (dry hopping), because they are rich in compounds that are responsible for the hop aroma.

## 5. Conclusions

In this research, aqueous extracts obtained from four varieties of American hops (Citra, Centennial, Chinook, and Mosaic) were analyzed. These analyses included the volatile fraction via GC-MS and sensory analysis via olfactory evaluation. 

In all samples, the main compounds identified were myrcene, β-caryophyllene, and α-humulene, terpene compounds that characterize the volatile fractions of hops. In addition, several compounds were found from the classes of terpenes, sesquiterpenes, alcohols, esters, ketones, and oxides. Furthermore, the sensory analysis allowed us to characterize the sensory profiles of each aqueous extract. Specifically, the Citra variety had an aroma with herbaceous and green fruit notes; Centennial was characterized by floral and citrus notes; the Chinook variety presented tropical fruit, pine, and resinous notes; and, finally, the Mosaic extract was mainly described by citrus aromas.

From a comparison with the literature, the aqueous extracts showed higher amounts of volatile compounds that characterize the hop aroma than those found in distillation extraction. This demonstrates that the use of hops aqueous extracts can provide sensory benefits to satisfy different consumer demands. 

In addition, their use would resolve the disadvantages of the use of pellets or powder hops during dry hopping, i.e., the high costs and the production of non-exhausted waste [55].

In conclusion, the future perspectives of research will be focused on the use of these extracts for the production of different beers (with different styles and characteristics) and on the improvement of the Synergy Pure technique.

## Figures and Tables

**Figure 1 foods-13-02454-f001:**
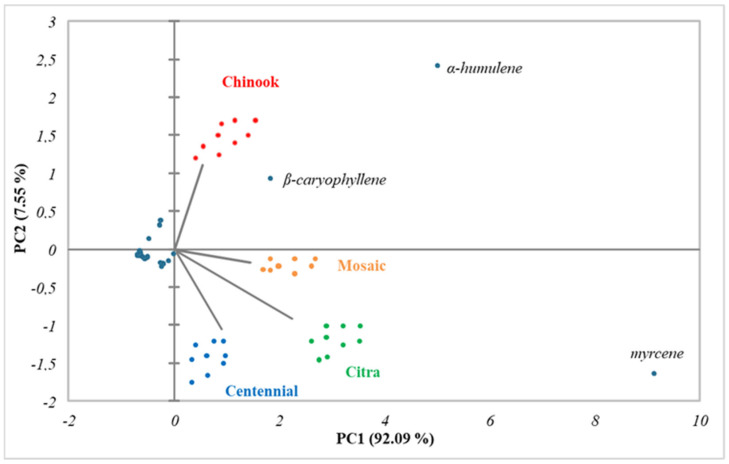
PCA score plot from the GS-MS analysis.

**Figure 2 foods-13-02454-f002:**
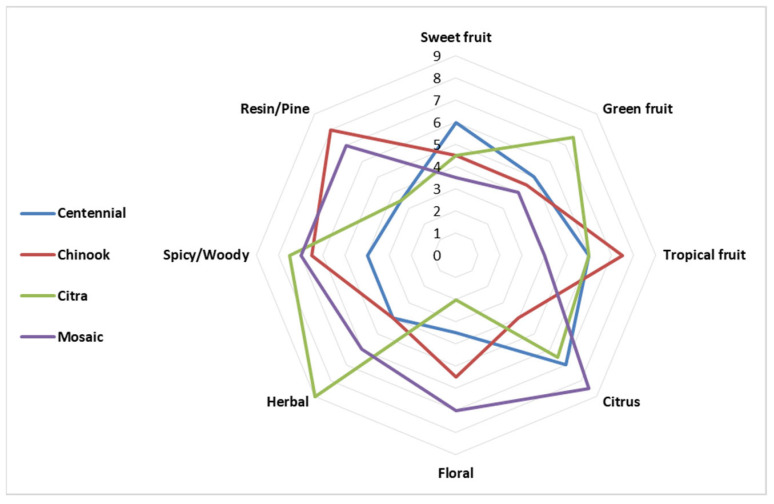
Spider plot representation of sensory analysis.

**Figure 3 foods-13-02454-f003:**
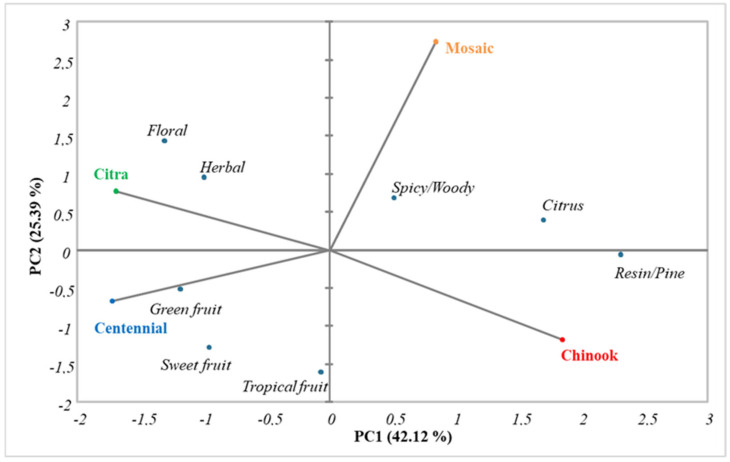
PCA score plot based on sensory analysis.

**Table 1 foods-13-02454-t001:** Average values of compounds identified and quantified (ppm) via GC-MS in extracts of Chinook, Centennial, Mosaic, and Citra hops.

Compound	RT (min) ^1^	LRI ^2^	Citra	Centennial	Chinook	Mosaic	Odor Descriptors ^3^
Dimethyl sulfide	1.28	765	- ^4^ a ^5^	0.02 ± 0.00 b	0.04 ± 0.00 b	-a	cabbage, sulfur, gasoline
α-Pinene	2.20	1022	4.66 ± 0.07 b	1.52 ± 0.03 a	1.73 ± 0.03 a	1.73 ± 0.02 a	pine, turpentine
Dimethyl disulfide	2.79	1074	0.93 ± 0.03 b	0.27 ± 0.00 a	0.90 ± 0.02 b	1.32 ± 0.04 c	onion, cabbage, putrid
Isobutyl alcohol	2.85	1089	0.30 ± 0.01	0.24 ± 0.00	0.50 ± 0.01	0.43 ± 0.02	wine, solvent, bitter
Isobutyl isobutyrate	2.91	1091	4.37 ± 0.05 b	1.92 ± 0.04 a	1.30 ± 0.02 a	4.01 ± 0.05 b	tropical fruit, pineapple
β-Pinene	3.12	1112	6.92 ± 0.06 b	9.71 ± 0.06 c	3.64 ± 0.03 a	2.04 ± 0.02 a	pine, resin, turpentine
Isoamyl acetate	3.40	1123	0.31 ± 0.01 a	0.15 ± 0.01 a	0.52 ± 0.01 b	0.23 ± 0.00 a	sweet fruit, banana
Myrcene	4.10	1163	945.71 ± 0.09 d	530.73 ± 0.10 c	274.42 ± 0.08 a	327.67 ± 0.05 b	fresh, balsamic
Isoamyl isobutyrate	4.84	1178	47.75 ± 0.08 c	25.32 ± 0.05 a	26.61 ± 0.03 a	31.78 ± 0.07 b	fruity, apple-like
p-Cymene	6.97	1270	1.07 ± 0.04 b	0.37 ± 0.01 a	0.21 ± 0.00 a	0.56 ± 0.01 a	solvent, gasoline, citrus
Isoamyl 2-methylbutyrate	7.34	1281	1.90 ± 0.03	1.84 ± 0.03	1.49 ± 0.01	1.23 ± 0.01	fruity, pineapple, strawberry
Methyl heptanoate	7.62	1289	1.72 ± 0.04 b	0.83 ± 0.03 a	0.36 ± 0.00 a	1.46 ± 0.04 b	sweet, slightly spicy, fruity
Isoamyl isovalerate	7.87	1291	2.09 ± 0.03 a	3.26 ± 0.02 b	5.11 ± 0.04 c	1.88 ± 0.06 a	sweet, fruity, green, soapy
1-Hexanol	9.78	1350	1.25 ± 0.05 b	0.30 ± 0.01 a	0.23 ± 0.01 a	2.05 ± 0.06 c	resin, flower, green
2-Nonanone	10.94	1385	2.22 ± 0.06 b	0.10 ± 0.00 a	0.23 ± 0.00 a	4.78 ± 0.09 c	hot milk, soap, green
Methyl octanoate	11.02	1388	4.64 ± 0.07 c	0.79 ± 0.01 a	0.43 ± 0.01 a	1.35 ± 0.03 b	green, floral, strong
Yalangene	13.66	1455	2.64 ± 0.04 b	0.87 ± 0.02 a	6.53 ± 0.07 c	1.01 ± 0.03 a	spicy, peppery
α-Cubebene	13.94	1462	9.43 ± 0.07 b	3.38 ± 0.02 a	24.93 ± 0.09 c	3.82 ± 0.02 a	herbaceous, wax
Methyl nonanoate	14.40	1496	5.70 ± 0.03 c	0.67 ± 0.00 a	0.83 ± 0.00 a	3.22 ± 0.01 b	fruity, tropical, coconut
Linalool	16.30	1544	47.84 ± 0.09 c	31.12 ± 0.07 b	17.67 ± 0.03 a	23.63 ± 0.05 a	flower, lavender, fresh notes
β-Caryophyllene	17.21	1588	205.68 ± 0.07 d	88.62 ± 0.05 b	181.58 ± 0.09 c	59.76 ± 0.04 a	wood, spice
2-Undecanone	17.72	1596	40.74 ± 0.07 b	2.65 ± 0.01 a	5.16 ± 0.02 a	34.02 ± 0.04 b	orange, fresh, green
Methyl 4-decenoate	18.63	1615	48.68 ± 0.06 d	23.29 ± 0.05 c	5.43 ± 0.02 a	18.35 ± 0.02 b	waxy, milky, green
α-Humulene	19.42	1667	380.78 ± 0.10 b	180.01 ± 0.03 a	418.03 ± 0.08 c	171.63 ± 0.09 a	woody, spicy
γ-Muurolene	20.12	1689	16.28 ± 0.04 b	9.59 ± 0.03 a	53.77 ± 0.13 c	6.74 ± 0.04 a	herb, wood, spice
Valencene	20.91	1726	39.05 ± 0.03 b	5.41 ± 0.02 a	48.37 ± 0.33 b	2.34 ± 0.01 a	green, oil, citrusy
2-Tridecanone	23.93	1785	8.37 ± 0.03 b	3.13 ± 0.01 a	2.21 ± 0.02 a	13.88 ± 0.04 c	fat, herbaceous, earthy
Geranyl isobutyrate	24.07	1792	6.60 ± 0.04 b	10.66 ± 0.07 c	2.63 ± 0.00 a	6.17 ± 0.02 b	sweet floral, fruity, green
Geraniol	25.12	1833	9.85 ± 0.02 a	52.23 ± 0.06 b	15.09 ± 0.04 a	13.10 ± 0.04 a	rose, geranium
β-Caryophyllene oxide	28.10	1981	14.37 ± 0.06 c	4.97 ± 0.08 b	1.64 ± 0.01 a	6.25 ± 0.05 b	herb, sweet, spice
Humulene epoxide	29.60	2044	16.45 ± 0.03 c	8.43 ± 0.02 b	5.50 ± 0.01 a	8.83 ± 0.05 b	spicy or woody
Octanoic acid	30.61	2061	1.61 ± 0.01 b	0.12 ± 0.00 a	0.10 ± 0.00 a	1.44 ± 0.02 b	sweat, cheese
Decanoic acid	35.68	2271	4.74 ± 0.02 c	0.67 ± 0.00 a	2.70 ± 0.02 b	1.26 ± 0.01 a	rancid, fat
All			1884.62 c	1003.20 b	1109.91 b	757.98 a	

^1^ Retention time. ^2^ Linear retention index. ^3^ Odor descriptors are based on the Flavornet Database (https://www.flavornet.org/index.html). ^4^ Not detected. ^5^ Different letters in the same row indicate significant differences at *p* < 0.05 among the samples.

## Data Availability

The original contributions presented in the study are included in the article, further inquiries can be directed to the corresponding author.

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
