# Peer review of "GC-MS and Sensory Analysis of Aqueous Extracts of Monovarietal American Hops, Produced Using the Synergy Pure™ Extraction Technique"

_foods, 2024, doi:10.3390/foods13152454_

Round 1

Reviewer 1 Report (Previous Reviewer 1)

Comments and Suggestions for Authors

Dear authors,

the manuscript has improved a lot due to the revision, which you have made. I have only few small details:

L. 173 - Distilled on a rotary evaporator? It doesn’t say anything. Do you mean, that you were evaporating the slurry until you have acquired 50 g of the ‘distillate’ (evaporated water with volatiles)? Or that you have evaporated the slurry until only 50g remained in the round bottom flask? Please, correct it, it doesn’t make a lot of sense.

L. 178 What is the internal standard? What is the concentration of the internal standard solution? Please, add it in that part of the manuscript, not only the GC-MS analysis, it makes more sense.

Author Response

Thank you very much for taking the time to review this manuscript. We are very happy that the new version of the manuscript was appreciated. Below are the responses to the comments point by point.

Comments 1: L. 173 Distilled on a rotary evaporator? It doesn’t say anything. Do you mean, that you were evaporating the slurry until you have acquired 50 g of the ‘distillate’  (evaporated water with volatiles)? Or that you have evaporated the slurry until only 50g remained in the round bottom flask? Please, correct it, it doesn’t make a lot of sense.
Response 1: More details have been reported on the production of aqueous extracts. In particular, the technique involved the use of a rotary evaporator. The slurry was distilled unit il a residue of 50.0 g was obtained at the bottom of the flask. This final slurry was used to prepare samples for the following analyses.

Comments 2: L. 178 What is the internal standard? What is the concentration of the internal standard solution? Please, add it in t hat part of the manuscript, not only the GC M S analysis, it makes more sense.
Response 2: The description of the internal standard has also been included in this paragraph. In particular, ethyl nonaoate was used at a concentration of 1 mg/mL. 

Reviewer 2 Report (Previous Reviewer 4)

Comments and Suggestions for Authors

Dear Authors,

I have revised the previous version of your manuscript and I can say that the new version is a better one. However, there are some things that should improved in it.

The title is too long and to some extent it is not correct. You can't say how your extracts will effect on beer quality without adding it in brewing.

Production of monovarietal Aqueous Extracts from American Hop using Synergy Pure™ extraction technique: a Focus on Brewing Quality and Aptitude Through GC-MS and Sensory  Analysis  ->  GC-MS and Sensory  Analysis of monovarietal Aqueous Extracts from American Hop, produced using Synergy Pure™ extraction technique 

Introduction

The Introduction is too long. There are things that should be removed as the detailed information for the names of the hop varieties, the thiols that were first studied in wines, etc. Other thing are the same as ln.49 and ln. 112. So, please re-write the Introduction.

I'm not sure that there is need of the second aim of your study because if you intend to use hop extract in brewing you definitely have to compared it with typical hop extracts.

Materials and Methods

There is no need of 2.8

According to me it will be better if you combine Results and discussion because it will be easier for the reader to understand which substances exactly are the main contributors to the hop extract flavour and aroma.

Section 4.3 is not for Discussion. It is for Result and dscussion. Moreover, you compare your results with the results of Duarte, but Duarte has investigated only Citra and Mosaic extracts, so be more punctual the results for myrcene for which hop variety are.

Author Response

Thank you very much for taking the time to review this manuscript. Happy to know that the new version of the manuscript has been appreciated. Requested actions will be taken into consideration to further improve the manuscript. Below are the responses to the comments point by point.

Comments 1: The title is too long and to some extent it is not correct. You can't say how your extracts will effect on beer quality without adding it in brewing. Production of monovarietal Aqueous Extracts from American Hop using Synergy Pure™ extraction technique: a Focus on Brewing Quality and Aptitude Through GC-MS and Sensory  Analysis  ->  GC-MS and Sensory  Analysis of monovarietal Aqueous Extracts from American Hop, produced using Synergy Pure™ extraction technique

Response 1: The title of the manuscript has been changed as follows: GC-MS and Sensory Analysis of Aqueous Extracts of monovarietal American Hops, produced using the Synergy Pure™ Extraction Technique.

Comments 2: Introduction The Introduction is too long. There are things that should be removed as the detailed information for the names of the hop varieties, the thiols that were first studied in wines, etc. Other thing are the same as ln.49 and ln. 112. So, please re-write the Introduction.

Response 2: The introduction was modified to make it shorter and focused on the topic of the manuscript. The main changes have been highlighted in red.

Comments 3: I'm not sure that there is need of the second aim of your study because if you intend to use hop extract in brewing you definitely have to compared it with typical hop extracts.

Response 3: We are aware that the composition of the extracts must be compared, but this is a preliminary study in which we want to characterise and then compare, and only on the basis of the results we will be able to prepare a further application study. Therefore, we think it necessary to specify the comparison made.

Comments 4: Materials and Methods There is no need of 2.8

Response 4: Section 2.8 aimed to emphasise the strategy used for the literature search. Bibliographic search showed very few works on a complete study of volatile fraction of hop extracts. However, we accept the observation and delete section 2.8.

Comments 5: According to me it will be better if you combine Results and discussion because it will be easier for the reader to understand which substances exactly are the main contributors to the hop extract flavour and aroma.

Response 5: Following the suggestion of the reviewer, Results and Discussion were combined. Section 4.3 moved to Results and Discussion. The Future perspective section has been separated from Results and Discussion.

Comments 6: Section 4.3 is not for Discussion. It is for Result and dscussion. Moreover, you compare your results with the results of Duarte, but Duarte has investigated only Citra and Mosaic extracts, so be more punctual the results for myrcene for which hop variety are.

Response 6: The section was modified detailing the comparison made with Duarte et al. Comparison with the Mosaic and Citra varieties was made only with data from the ultrasound extraction with hexane. In fact, Duarte et al. reported only average values of volatile compounds extracted by distillation. Unfortunately, as also stated in the manuscript, the literature is poor in studies analysing the volatile fraction, especially in its complexity, of hop extracts.

Round 2

Reviewer 2 Report (Previous Reviewer 4)

Comments and Suggestions for Authors

Dear Authors,

Thank you for taking into account all my proposals! You have to correct only one thing:

3. Results -> 3. Results and discussion

This manuscript is a resubmission of an earlier submission. The following is a list of the peer review reports and author responses from that submission.

Round 1

Reviewer 1 Report

Comments and Suggestions for Authors

Dear authors

The manuscript is very hard to read, many of the sentences are translated in such a way, that they do not make any sense. I highly recommend getting the help of someone more proficient in the English language to rewrite the article.

The methodology seems all right, while the goal of the study is quite novel, but not revolutionary. It would be good to determine usefulness of these extracts in the production of beer, but it can be the objective of another study.

Below are more specific comments:

Firstly - abstract should be rewritten, it looks like it was written hastily at the end of the work.

L. 28-31 Italics for latin words

L. 34-37 Sentence is unreadable. What components? Are these components about 4 percent of polyphenols? How are components from the group of polyphenols related to aroma? Please, rewrite the sentence.

L. 38 Contain, not include. Also, it is not true that alpha acids indirectly contribute to the bitter aroma. Additionally, bitter is a taste, flavour, not aroma. Also, beta-acids also have bitter flavour, but the iso-alpha-acids are, in fact, the major contributors to the bitter taste (not aroma!) of the beverage.

L. 43-44 While this sentence is true, nothing from the previous part of introduction suggests this outcome.

L. 45-48 Thanks to GC methods 1000 compounds are assumed to be present? Doesn’t make sense. Rewrite.

L. 49-52 And, very importantly, weather. Temperature, amount of rainfall and sunlight play crucial part in the composition of essential oils.

L. 54 Analytical analysis?

L. 57-59 I think that saying that the research in the area of hop essential oil was mainly focusing on the finding of the varietal aromatic differences of each hop variety is a little bit stretched.

L. 61 ‘Might be’ or ‘are’ involved?

L. 66 Linalool as a main indicator of hop aroma? What about myrcene and humulene?

L. 86 L is the abbreviation for the liter. And it is always beter to use dm³.

L. 98-100 The sentence makes little to no sense.

L. 107-109 This profile is different than traditional, European varieties, but dozens of hop varieties from USA or New Zealand are characterised with similar odour, which is why claim, Citra is ‘one and only’ variety with citrusy aromas. It just isn’t true.

L. 109-110 A reference for that, perhaps?

L. 116 Aroma of a barrel?

Plant matrix - cultivars were purchased? Or T90 hop pellets were purchased? Please, add the year of the harvest.

Standards - ethyl-nonanoate used as an interal standard? In what solution?

L. 131-152 Some references?

L. 166 ‘Temperatures below 100 degrees’ is a rather broad spectrum.

L. 187 Please, use metric units, not ‘psi’.

L. 187-190 Please, add the total run time.

L. 196-197 You have written that you have used standards, but in the section about the standards you only give the internal standard. Why?

L. 233 Walls per million?

Table 1 - why are the standard deviations absent? Odour descriptors are based on…? Also, units should be given at the top of the table.

L. 258 / L. 265 What authors?

L. 273-275 Please, give the threshold.

L. 325 Biplot.

L. 378-379 Some source?

Comments on the Quality of English Language

The manuscript is very hard to read, many of the sentences are translated in such a way, that they do not make any sense. I highly recommend getting the help of someone more proficient in the English language to rewrite the article.

Reviewer 2 Report

Comments and Suggestions for Authors

Note: The problems listed are representative, not exhaustive, and many other relevant issues need to be addressed.

1. Abstract: “......four American hop varieties......”. It is recommended to add the names of the varieties in detail to make it more readable. 

2. Preface: the present one is too long and lacks scientific presentation. The author describes a lot about the varieties and aromas of hops. The question is: On the one hand, what is the status of the use of hops in brewing? What limitations exist? Based on the shortcomings of existing studies, the superiority and necessity of the “Synergy Pure™ extraction technique” could be better demonstrated. On the other hand, the purpose of this paper is not clear, and what are the innovations compared to the previous studies? Finally, the abstract mentions “hop creep”, and it is suggested that the introduction should be supplemented with relevant background information.

3. Sections 2.3-2.4 are confusing about the injection method of GC-MS. “1 mL of supernatant was transferred into 1.5 mL gas chromatography vial”, then how did it enter the GC-MS system for analysis? If it was a liquid injection, what was the injection volume?

4. section 2.5, the description of the tasting panel is suggested to be refined, age/gender/training process ......

5. 3 biological replicates are not reflected in the data in Table 1. In addition, the calculation of retention index (RI), which is generally indispensable for qualitative work on volatile compounds, is also not reflected in the authors’ work. It gives reasons to doubt the authenticity of the data.

6. Figures 1 and 3 are unclear.

7. Overall, the experimental design is very simple, and the depth of discussion is insufficient and does not meet the basic requirements of a scientific paper.

Comments on the Quality of English Language

Extensive editing of the English language required.

Reviewer 3 Report

Comments and Suggestions for Authors

The article title - "Sensory Analysis and GC-MS of Aqueous Extracts of American Hop Varieties" authored by G. Tripodi, M. Zaninelli, A. Cappelli, M. Ferluga and G. Dima reports detection of volatile compounds and sensory properties of American hop verities, including: Citra, Centennial, Chinook and Mosaic, using GC-MS technique and Sensory Analysis.

The overall writing quality of the article is satisfactory.

The experimental section explains the sample analysis technique. 

The data accuracy seems to be oaky.

However, the scientific problem/question of this study is not fully understood. 

American hop verities: Citra, Centennial, Chinook and Mosaic, are already in use to develop beers in commercial scale. Therefore, compatibility of the physical, chemical and sensory properties of those hop verities to produce beer are well established.

Do the authors suggest that those physical, chemical and sensory properties of American hop verities: Citra, Centennial, Chinook and Mosaic studied and reported first time in this study? If so, they must mention this in the text. If not, they should compare their experimental results with published/available parameters.

In Table 1, it is not clear whether the experimental numbers are average of three readings. If those are average of three readings, standard deviation of the readings should be reported.

The Discussion sections #3.1 and #3.2 can be improved using sub-headings.

Section#4 is missing in the article (I presume Conclusion would be Section#4).

This article reports strong experimental results of detecting physical, chemical and sensory properties of American Hop verities using GC-MS technique and Sensory analysis. However, the critical question of the study is missing. Authors must address why this study is required for aforementioned Hop verities which are used for commercial beer production.

Please address the above queries/comments. 

Comments on the Quality of English Language

The quality of English is okay.

Reviewer 4 Report

Comments and Suggestions for Authors

Dear Authors,

The topic of the manuscript is interesting but a lot of things should be improved.

Introduction

It is too long and a lot of things that are written in Introduction were written again in Results and discussion (e.g. ln. 66-72 and ln. 248-254).

Latin names of plants shoild be written in Italic.

ln. 35-36 percent -> %

ln. 87 4,2 ng/l -> 4.2 ng/l

ln. 132-144 It will be better if you move it to Material and methods or Results and discussion

Results and discussion

ln. 233 walls per million?

Conclusion

ln. 407 destrins?

In Conclusion you have mentioned the advantages of using your hop extracts but everything has disadvantages, so what are the disadvantages of this hop extracts? Have you tried to make beer with them and have you compared it to the conventional or dry hopped beer?

Round 2

Reviewer 1 Report

Comments and Suggestions for Authors

The manuscript is significantly improved. However, I will recommend ‘major revision’, even though I think that manuscript should undergo ‘minor revision’, because by suggesting ‘minor revision’ the manuscript could be accepted without the need of consulting the reviewers and I really think that you should describe the GC-MS method properly and without mistakes for the analysis to be repeatable.

Hope you understand.

L. 9 Cross out the ‘properties’

L. 10 Change ‘appropriateness’ to ‘viability’ or ‘possibility’

L. 177 mL? Impossible. Check the unit.

L. 178 You have changed the ‘psi’ to mL/min. Therefore, you have described a new method. In the GC analysis, you use constant pressure or constant flow. In the first version of manuscript, you have described, that you have used constant pressure (therefore, changing flow) and in the second version you describe that the flow is constant (therefore, the pressure is changing). Which one was the method used? In the last review I suggested you not to use psi, but metric measurement, but of pressure (like kPa) and not changing the description of the method. Please, confirm which method was used with the GC-MS operator (I assume it was not the author who wrote the manuscript) and decide, which version is proper.

Reviewer 2 Report

Comments and Suggestions for Authors

The introduction is still confusing, and I admit that the authors have adequately described the flavor studies of hops. But, more importantly, what is the reason for the switch from “hops” to “hops aqueous extract”? What are the operational methods of “hops extracts”? Does the loss of bitterness affect the taste of the final product as “hops extract does not contain resins”? etc. In addition, the experimental design of the article is too simple and the results presented are not meaningful.

Comments on the Quality of English Language

Moderate editing of English language required.

Reviewer 4 Report

Comments and Suggestions for Authors

Dear Authors,

You have taken into account some of my suggestions and it has affected positively the manuscript quality.